# Effects of a Complex Physical Activity Program on Children's Arithmetic Problem Solving and Arithmetic Reasoning Abilities

**Gianpiero Greco** [1], **Luca Poli** [1], **Roberto Carvutto** [1], **Antonino Patti** [2], **Francesco Fischetti** [1,*]
and **Stefania Cataldi** [1]

[1] Department of Translational Biomedicine and Neuroscience (DiBraiN), University of Study of Bari, 70124 Bari, Italy
[2] Sport and Exercise Sciences Research Unit, Department of Psychology, Educational Science and Human Movement, University of Palermo, 90144 Palermo, Italy
*  Correspondence: francesco.fischetti@uniba.it

**Abstract:** Research has shown that higher levels of physical activity are associated with better cognitive performance in children. However, the benefits of physical activity on academic achievement and specifically on mathematics performance need to be further explored. Therefore, this study aimed to investigate the effects of a complex physical activity (CPA) intervention program, including cognitive involvement, on children's mathematics performance. The participants were 128 children (aged 12–13 years) attending third grade in three middle schools. They were randomly allocated into a CPA intervention (n = 64) or a waitlist control group (n = 64), the latter of which was given a regular and easy-to-perform activity program. At baseline and after the intervention, students' physical fitness was measured using a battery of standardized motor tests (20 m shuttle run test, curl-up test, push-up test, and sit and reach test). In addition, the AC-MT 11-14 test was administered to allow a standardized and comprehensive assessment of arithmetic problem-solving and arithmetic reasoning abilities. In comparison to the control, the intervention group showed significant improvements ($p < 0.001$) in comprehension and production ($d = 1.88$), arithmetic reasoning ($d = 2.50$), and problem solving ($d = 1.32$), as well as in 20 m shuttle run test ($d = 1.81$), push-up test ($d = 1.88$), curl-up ($d = 3.52$), and sit and reach ($d = 2.52$). No significant changes were found in the control group. In conclusion, findings showed that a 12-week CPA intervention program can improve mathematical performance in children in the third grade of middle school. CPA intervention may be an effective method to improve academic performance and avoid student failure.

**Keywords:** cognitive function; academic performance; physical exercise; school-based physical activity

## 1. Introduction

High-quality educational strategies play a significant role in preparing students to achieve their academic outcomes [1–3], especially when it comes to mathematical competence [4]. In the last decade, a large body of research has been carried out on children's mathematical disabilities since the consequences of poorly developed mathematical competencies affect employment and day-to-day living in the modern world [5–7]. Why, then, is it extremely difficult for certain students to perform arithmetic operations or solve a mathematical problem? The mathematical area consists of different domains that are partly independent and separable. Indeed, an essential distinction concerns mathematical calculation disorders and arithmetic problem-solving and reasoning abilities disorders [8,9]. Both are related to disorders that concern multiple functions, such as working memory, attention, and metacognition. Individuals with difficulties in arithmetic problem-solving and arithmetic reasoning abilities may have adequate abilities in the management of numbers and calculation procedures, but they present difficulties in the identification and integration

of verbal information, visuospatial representation, and phase planning of the necessary procedures to solve the problem [10].

Arithmetic problem solving and reasoning require a series of complex cognitive skills and abilities [11], such as (i) coding, which is the ability to translate the information obtained from the text to build an internal representation of it. Language competence and semantic memory contribute to carrying out this process. However, a process of cognitive integration is necessary to put these steps together: (ii) solving, which consists of searching in memory for the necessary information to reach the final solution; (iii) forecasting, namely predicting whether it is possible to solve the problem; (iv) planning, which is the development of a project; (v) monitoring, which allows control of the process; and (vi) appraisal, which means evaluating the result.

A potential way to promote arithmetic problem solving and arithmetic reasoning is through a school-based physical activity (PA) program [12]. In this regard, in addition to the positive effect of PA on physical fitness, a growing interest has emerged in studying its potential influence on children's cognitive domain, as well as on academic achievement [13–16]. Specifically, the PA that affects arithmetic problem-solving and reasoning abilities is complex physical activity (CPA) [17,18], namely those forms of PA intervention with cognitive involvement through complex motor coordination in which the activities require the individual's abilities to control and adapt movements, to focus their abilities, and to recall movements previously learned to make ensuing movements [19]. It has been demonstrated that at least six weeks of CPA intervention could be enough to improve mathematical performance [17].

Concerning the relationship between CPA and learning, several studies demonstrate how CPA impacts not only the most elementary learning process but also those more complex ones, such as the understanding of a text and the solution to a problem [20]. Various mediating factors may promote the impact of CPA on academic outcomes among schoolchildren. The latest research indicates that CPA fosters several changes in the structure and function of the brain, such as energy metabolism and synaptic plasticity [21], and may promote children's development through positive effects on brain systems that underpin cognitive functions and behavioral attitudes [22]. Moreover, there is significant evidence indicating that CPA during the school day may positively affect children's cognitive and behavioral school engagement [23], attention, time-on-task [24], and executive functions [25], which are associated with achievement in both reading and mathematics [26]. A different mediating factor emerges from the evidence whereby children's motor development is related to cognitive learning to such an extent that CPA may affect academic performance [27]. After all, several studies showed that many cognitive skills, including visuospatial skills [28], rapid automatized naming, processing speed, and memory skills, are influenced by those physical activities that implicate cognitively engaging tasks [29].

There has been considerable research into physical activity interventions in the educational context, but little work has explored how a CPA program could affect arithmetic problem-solving and reasoning abilities disorders. Therefore, the present study aimed to investigate the effects of a 12-week CPA intervention program on children's mathematical performance.

## 2. Materials and Methods

### 2.1. Study Design

A randomized controlled study design was used to investigate the effects of a 12-week CPA program on children's arithmetic problem-solving and arithmetic reasoning abilities. In a municipality in the southern part of Italy, 128 students attending the third grade in three middle schools were randomly allocated into a CPA intervention (n = 64) or a waitlist control group (n = 64), the latter of which was given a regular and easy-to-perform physical activity program. Participants were pair-matched based on gender, and randomization was performed by Research Randomizer, a program published on a publicly accessible official website (www.randomizer.org, accessed on 1 February 2022). The researchers (both physical education teachers) were blinded to this intervention randomization and the

control group allocations. Both the training programs were performed for 60 min, 2 days per week, for a total of 12 weeks. This protocol was carried out during PA curricular hours. Measurements were administered one week before the intervention (pre-test) and after the intervention (post-test).

*2.2. Participants*

One hundred twenty-eight students (M age = 12.85, SD = 0.36, years) from six third grade classes of three local middle schools were voluntarily recruited to participate in the study. The following inclusion criteria were identified to constitute a convenient sample that could respond to the study needs: children from 12 to 13 years of age, attending the chosen schools, being relatively healthy individuals, with no learning difficulties or disabilities, capable of completing an exercise session, and able to abstain from other physical activities outside the study protocol.

To establish the sample size needed for the study, an a priori power analysis [30] with an assumed type I error of 0.05 and a type II error rate of 0.05 (95% statistical power) was calculated and revealed that 54 participants in total would be sufficient to observe medium "time × group" interaction effects ($f = 0.25$). However, to account for possible drop-out, larger samples were recruited.

According to the inclusion criteria, 135 subjects were invited to participate in the study. Of those, 7 declined to participate due to personal reasons, and 128 students agreed to be in the research study and completed the baseline measurements. Consequently, the final sample consisted of 128 participants, which were matched and randomly assigned to one of two treatment conditions. The experimental group (n = 64) was composed of 34 males and 30 females, while the wait list control group (n = 64) involved 33 males and 31 females. Principals were informed about the study. Upon principals' approval, written informed consent was secured from all parents of the participants, and they were informed that they could withdraw it at any time. This study was conducted in accordance with the Declaration of Helsinki and approved by the Ethics Committee of Bari University (protocol code 0024425 | 11/03/22).

*2.3. Procedures*

The intervention program was carried out during PA curricular hours in the school gym. Standardized motor assessment tests and mathematical performance were assessed in the mornings during the mandatory physical education lessons, in the week before and the week after the intervention period, to determine the participant's starting level and to detect if any changes occurred compared to baseline.

The participants completed each test at the same time of the day and under the same experimental conditions. Students were tested individually, and each task was explained before the participants started. Children were not aware of the purpose of the study or the experimental conditions to avoid any possible effect that may alter the accuracy of the data. Participants were instructed to wear suitable sportswear to limit possible variability within the testing procedure and were instructed to avoid excessive physical exertion 24 h before each testing session. All measurements for testing and both intervention programs were instructed, supervised, and performed by two experienced physical education teachers, certified by the Italian Ministry of Education. The reliability of the dependent measures was calculated using the intraclass correlation coefficient (ICC). To measure test-retest reliability, we conducted the same test twice on the entire sample using the two best trials among the three recorded in the pre-test; therefore, we calculated the correlation between the two sets of results as follows: ICC = (variance of interest)/(total variance) = (variance of interest)/(variance of interest + unwanted variance). A value of the ICC below 0.5 indicates poor reliability, between 0.5 and 0.75 moderate reliability, between 0.75 and 0.9 good reliability, and above 0.9 excellent reliability [31].

*2.4. Measures*

2.4.1. Motor Tests

Students' physical fitness was measured using a battery of standardized motor tests, specifically the 20 m shuttle run test, curl-up test, push-up test, and sit and reach test.

The 20 m shuttle run test [32] was used to estimate aerobic fitness. The running speed was increased gradually, every 1 min after the signal of start, until maximum voluntary exhaustion. The score of the test was the result of the number of laps run. Each participant performed three trials, and the best score between the trials was recorded. The test-retest reliability reported high reliability for the 20 m shuttle run (ICC (r)= 0.96).

Muscular fitness was determined via a curl-up test for abdominal strength [33], and a push-up test [34] for upper-body muscle strength. In the first instance, the score was determined by the maximal number of repetitions, while in the second, the result was the number of push-ups performed during 1 min. Furthermore, the push-up test involved slightly different techniques, namely, boys had their hands and toes on the ground, while girls had their hands and knees on the ground. Each participant performed three trials for both tests, and the best score achieved by the three measures was recorded. The test-retest reliability reported high reliability for the curl-up test (ICC (r)= 0.99) and the push-up test (ICC (r)= 0.93).

Lastly, the sit and reach test [35], a common measure of flexibility of the lower back and hamstring muscles, required the collection of three measures, and the score was the best reached by the three distances. The test-retest reliability reported high reliability for the sit and reach test (ICC (r)= 0.91).

2.4.2. AC-MT 11-14 Test

The AC-MT 11-14 [36] is a well-validated test that allows a standardized and comprehensive assessment of arithmetic problem-solving and arithmetic reasoning abilities. It consists of two specific sections: the collective part, on numbers and calculation, and problem-solving. They required collective administration and concerned knowledge of numbers, arithmetic reasoning, and problem solving. Regarding the first section, it is composed of eight subtests (perform the operations, arithmetic expressions, what's the biggest, transform into numbers, complete the series, transcribe in numbers, approximate calculation, facts, procedures, and principles), and three macro-variables can be identified, namely written calculation, comprehension and production, and arithmetic reasoning.

The test takes about 60 min for the first section and 30 for the second (including the instruction and practice phase). The scoring system assigns 1 point for each correct answer and 0 points for each wrong answer.

*2.5. Intervention Program*

The exercise training program for the experimental group provided a variety of complex motor movements that require the students' ability to control and adapt movements, focus their abilities, and to recall movements previously learned to make ensuing movements. Within these exercises, different complex coordinative abilities were stressed through the ability to maintain balance, react to stimuli, adjust, and differentiate movements. Specifically, the activities were organized in various types of circuits, made up of several stations, that stimulate motor skills with different running, throwing, catching, and jumping movements, which were combined and matched in a complicated manner. Furthermore, several obstacles to overcome and different goals to be achieved were added. Each training begins with 10 min of warm-up, continues with 40 min of core training, and ends with 10 min of cool-down. The control classes were offered an equal amount of simple, easy-to-perform physical activity programs, consisting of a warm-up and cool-down phase identical to the CPA program and a core phase characterized by walking, running, jumping, skipping, and animal-like walking exercises. The CPA intervention program is described in Table 1.

**Table 1.** Complex physical activity intervention program.

| Phase | Exercise | Guidelines |
|---|---|---|
| **Warm-up** | Marching in place, wide toe touch, leg swings, arm swings, half jacks, chest expansions, torso rotation, alt back expansions, shoulder rotations, hops on the spots, single-leg hops, hip rotations, walking jacks, hip circles, walking knee hugs, side shuffles. | Duration: 10 min. Perform each exercise for about 60 s, 1 set. |
| **Core training** | Jumping down the bench<br>Dribbling a tennis ball<br>Standing balance with ball tosses<br>Reacting side squats<br>Facing the hoops with monopodalic alternated backers<br>Facing the stakes with sides slipping<br>Jumping mini hurdles and high hurdles; low hurdles must be jumped up at maximum speed, while the high ones must be jumped while going slowly<br>Running with high knee raises<br>Passing the cones with changes of direction<br>Facing the mini hurdles with elongated steps<br>Running and controlling a soccer ball with the foot<br>Bouncing a volleyball alternating with the left or right hand while standing on a bench<br>Throwing a handball alternating between the left and right hand into a gymnastic hoop<br>Bouncing a basketball and a volleyball, respectively, with the left and the right hand at the same time<br>Hitting with the head the ball launched by the teacher | Duration: 40 min. Training load: 1–2 sets of 8–15 repetitions with 45 sec of slow walking between each exercise. Progression: increase repetitions before sets. |
| **Cool-down** | Glute stretch, standing quad stretch, piriformis stretch, side bench stretch, arm-cross shoulder stretch, overhead triceps stretch, lower back stretch, abdominal stretch, lunge with spinal twist, butterfly stretch, seated shoulder squeeze, child pose, breathing exercises. | Total duration: 10 min. Overload: stretch beyond resting length but not beyond pain-free ROM. Duration: 10–30 sec/stretch; repetitions: 2–4; accumulate 60 sec per exercise. Progression: gradual increase in stretch duration or repetitions. |

*2.6. Statistical Analyses*

Statistical analyses of the outcomes were conducted using IBM SPSS Statistics, version 26.0 (2019 SPSS Inc., IBM Company, New York, NY, USA). Data were presented as group mean (M) values and standard deviations (SD) and were checked for assumptions of homogeneity of variances via Levene's test. The Shapiro–Wilk test was used to test the normality of all variables. An independent sample *t*-test was applied to detect any group differences at baseline, and then a two-way ANOVA (experimental/control group) x time (pre/post-intervention) with repeated measures was performed to analyze the effect of the training on all examined variables. Subsequently, when "group x time" interactions showed significance, group-specific post hoc tests (paired *t*-test) were conducted to identify the significant comparisons. Partial eta squared ($\eta^2_p$) was used to estimate the magnitude of the difference within each group and defined as follows: small: $\eta^2_p < 0.06$, medium: $0.06 \le \eta^2_p < 0.14$, large: $\eta^2_p \ge 0.14$. In addition, Cohen's *d* was calculated for the post hoc tests. The criteria to interpret the magnitude of the Cohen's *d* were as follows: *d* = 0.20, small effect size; *d* = 0.50, medium effect size; and *d* = 0.80, large effect size [37]. Statistical significance was set at $p \le 0.05$.

### 3. Results

The two groups of participants received the treatment conditions as allocated, and their average adherence (attendance) to intervention sessions was 99.16 % (23.8 of 24 actual sessions). There were no injuries resulting from the CPA intervention. The experimental and control groups did not differ at baseline in motor and mathematical performance ($p > 0.05$). Changes after the 12-week CPA intervention program are shown in Table 2.

**Table 2.** Changes after 12-week complex physical activity intervention program.

| | Experimental Group (n = 64) | | | Control Group (n = 64) | | |
|---|---|---|---|---|---|---|
| | Baseline | Post-Test | Δ | Baseline | Post-Test | Δ |
| **Motor Test** | | | | | | |
| 20 m shuttle run test (n) | 5.98 (1.98) | 8.45 (1.41) †* | 2.46 (1.35) | 6.65 (1.46) | 5.57 (1.41) | −1.07 (0.76) |
| Push-up test (reps) | 8.45 (2.09) | 11.84 (2.76) †* | 3.39 (1.79) | 8.42 (1.97) | 6.89 (1.96) | −1.53 (1.28) |
| Curl-up test (reps) | 29.04 (2.00) | 31.56 (2.12) †* | 2.51 (0.71) | 29.31 (2.21) | 27.35 (1.82) | −1.95 (1.22) |
| Sit and reach test (cm) | 18.95 (2.01) | 21.03 (2.44) †* | 2.07 (0.82) | 19.67 (1.86) | 17.51(2.46) | −2.15 (2.19) |
| **AC-MT 11-14 Test (score)** | | | | | | |
| Written calculation | 5.56 (2.17) | 5.34 (2.31) | −0.21 (0.48) | 6.17 (2.40) | 5.06 (2.11) | −1.10 (1.60) |
| Comprehension and production | 14.09 (3.24) | 15.85 (2.98) †* | 1.76 (0.93) | 14.60 (2.75) | 13.57 (2.59) | −1.03 (1.06) |
| Arithmetic reasoning | 21.43 (3.13) | 23.43 (3.31) †* | 2.00 (0.79) | 22.04 (3.07) | 21.37 (3.15) | −0.67 (1.53) |
| Problem solving | 6.14 (2.12) | 7.48 (1.79) †* | 1.34 (1.01) | 6.17 (2.34) | 4.39 (1.78) | −1.78 (1.80) |

Notes: values are presented as mean (±SD); Δ: pre- to post-training changes; † significant "group x time" interaction: a significant effect of the intervention ($p < 0.001$). * Significantly different from pre-test ($p < 0.001$). n = number of shuttles; reps = number of repetitions; cm = centimeters.

#### 3.1. Motor Tests

A two-factor repeated measures ANOVA found a significant "time x group" interaction for the 20 m shuttle run test ($F_{1,126}$ = 332.51, $p < 0.001$, $\eta^2_p$ = 0.72, large effect size), push-up test ($F_{1,126}$ = 317.98, $p < 0.001$, $\eta^2_p$ = 0.71, large effect size), curl-up ($F_{1,126}$ = 634.78, $p < 0.001$, $\eta^2_p$ = 0.83, large effect size), and sit and reach test ($F_{1,126}$ = 208.31, $p < 0.001$, $\eta^2_p$ = 0.62, large effect size). Post hoc analysis revealed that the experimental group made a significant increase from pre- to post-test in the 20 m shuttle run test ($t$ = 14.55, $p < 0.001$, $d$ = 1.81, large effect size) and an increase in the push-up test ($t$ = 15.10, $p < 0.001$, $d$ = 1.88, large effect size), curl-up ($t$ = 28.24, $p < 0.001$, $d$ = 3.52, large effect size), and sit and reach test ($t$ = 20.21, $p < 0.001$, $d$ = 2.52, large effect size). No significant changes were found for the control group ($p > 0.05$).

#### 3.2. AC-MT 11-14 Test

Statistical analysis showed significant "time x group" interaction for AC-MT 11-16 test in three of four macro-variables, namely comprehension and production ($F_{1,126}$ = 247.52, $p < 0.001$, $\eta^2_p$ = 0.66, large effect size), arithmetic reasoning ($F_{1,126}$ = 153.01, $p < 0.001$, $\eta^2_p$ = 0.54, large effect size), and problem solving ($F_{1,126}$ = 146.17, $p < 0.001$, $\eta^2_p$ = 0.53, large effect size). The post hoc analysis revealed a significant improvement in the experimental group score for comprehension and production ($t$ = 15.05, $p < 0.001$, $d$ = 1.88, large effect size), arithmetic reasoning ($t$ = 20.08, $p < 0.001$, $d$ = 2.50, large effect size), and problem solving ($t$ = 10.62, $p < 0.001$, $d$ = 1.32, large effect size). No significant changes were found for the control group ($p > 0.05$).

### 4. Discussion

The present study aimed to examine the effectiveness of a CPA program on arithmetic problem-solving and reasoning abilities disorders among third-grade middle school children, compared to traditional lessons. Results showed that performing a CPA program had a positive effect on mathematical performance among third-grade middle school students. This effect was stronger for the CPA program compared to traditional lessons where simple physical training was performed, which did not provide any cognitive involvement.

Confirming our hypotheses, the effects were higher in the CPA group, which performed physical activities with greater cognitive involvement, than the mere simple exercises carried out by the control groups. Therefore, this study seems to be consistent with previous literature that holds that PA that requires a cognitive effort has beneficial effects on educational outcomes through the introduction of complex physical exercise [38,39]. Indeed, this research, comparing the CPA program with regular physical education lessons, has found improvements in three of the four considered variables, namely comprehension and production, arithmetic reasoning, and problem solving. Thus, this study showed that CPA could increase the ability to perform arithmetic problem solving and arithmetic reasoning. We believe that this may be explained by the fact that the CPA program provided in this research required more of the child's ability to concentrate on the task, greater capacity to focus on the target, and rapid reaction prowess [40,41]. While the simple physical exercise given in this study required the students to simply remember the previously acquired information and relate it to the next movement and task, the repetitions of movements were more important than the ability to involve in-depth cognitive functions [42,43]. Several researchers claim that certain cognitive processes such as memory, attention, psychomotor, and visual-perceptual factors, all domains that could be affected by physical activity with cognitively engaging tasks, play a crucial role in the origin of a child's difficulties in the mathematical field [44–46].

However, the CPA program has not been able to improve the written calculation variable in children. The lack of influence of complex physical activity on this domain may be due to the dissociation between skills related to calculation and skills such as arithmetic reasoning and problem solving [47]. Indeed, an insufficient ability of calculation often depends on a lack of essential knowledge necessary to proceed in the subsequent learnings, and on the previous competence that students have acquired and built up during their educational path [48]. Quite the opposite, to develop adequate arithmetic reasoning, a series of complex cognitive skills is required, which are not necessary for the calculation capability [49]. Behind the difficulties in arithmetic reasoning and problem solving would seem to be a serious difficulty in the elaboration and conceptualization of the number (reading and writing of the numbers, assessment of the numerousness or quantity). In this regard, a severe difficulty arises in the recovery of memory (so-called "numerical facts"), and in carrying out the necessary procedures to perform written and mental calculations [50]. This explains the many cases of prodigious individuals who can perform very complex mental calculations but, at the same time, have a poor reasoning ability. It is not uncommon to find these amazing calculation abilities in individuals with a level of intelligence that is lacking, with serious cognitive and social incompetence, or with conditions such as autism. Therefore, these findings were in line with the previous physiological mechanisms hypothesis, whereby complex physical activity may elicit changes occurring in children's brain function that improve cognitive function due to greater cerebral blood flow, which affects learning, memory, and the ability to focus on certain tasks and conditions [51,52]. We believe that the CPA program proposed in this study helped to develop students' motor skills, which in turn allowed them to support other cognitive structures, growing the allocation of attitudinal resources and the efficiency of neurocognitive processes, through the ability to explore and understand the proposed tasks [53,54]. Accordingly, a 12-week CPA intervention that guided middle school children through movements with cognitively engaging tasks showed significant differences in mathematical performance compared to controls, which remained unaffected after training.

However, despite the direct and positive relationship between complex physical activity and mathematical outcomes, some limitations were present within this study. Most importantly, the first limitation was related to the relatively small sample size, which generated difficulties in recruiting motivated students or students who were able to abstain from other physical activities outside the intervention protocol during the training period to participate. The second limitation was that students were recruited from two local schools placed in a small district; thus, the results may not be generalizable to children at

other institutions or with other demographics. Lastly, we did not evaluate the long-term effects of CPA on cognitive skills, and the impact of other variables that might also affect mathematical performance, such as motivation, anxiety, interest, and IQ of children, have not been examined. Nevertheless, the strengths of this study were represented by the contribution that this effective approach brings to mathematical performance, as well as physical and cognitive health, among students. Therefore, because of the above, schools should provide more opportunities to influence children's health, physical fitness, and cognitive performance.

## 5. Conclusions

Physical activity provides several beneficial psycho-physical effects. Specifically, a CPA intervention that involved cognitively engaging movements may be one of the stimulation models for the development of better mathematical outcomes among school children. Considering this, schools should provide more opportunities to promote children's health, physical fitness, and cognitive performance. Otherwise, sacrificing physical education classes for other curricular subjects would not improve academic performance; on the contrary, it would lead to children having poor motor and cognitive skills. Nevertheless, more studies are needed to comprehend the effect of different movement strategies on academic performance, specifically in mathematics, in depth. Based on the results of research and discussion, it can be concluded that a CPA intervention that implicated middle school children through movements with cognitively engaging tasks may be a significant approach to improve academic achievement and avoid student failure.

**Author Contributions:** Conceptualization, G.G. and S.C.; methodology, G.G. and F.F.; software, G.G.; validation, G.G., L.P., and R.C.; formal analysis, A.P.; investigation, L.P.; resources, R.C.; data curation, G.G.; writing—original draft preparation, G.G. and L.P.; writing—review and editing, G.G., S.C., and F.F.; visualization, R.C. and A.P.; supervision, G.G., F.F., and S.C. All authors have read and agreed to the published version of the manuscript.

**Funding:** This research received no external funding.

**Institutional Review Board Statement:** This study was conducted in accordance with the Declaration of Helsinki and approved by the Ethics Committee of the Bari University (protocol code 0024425 | 11/03/22).

**Informed Consent Statement:** Informed consent was obtained from all parents of the participants involved in the study.

**Data Availability Statement:** The data presented in this study are available on request from the first author.

**Acknowledgments:** We thank the children who participated in the study, the parents, and the principals for their availability.

**Conflicts of Interest:** The authors declare no conflict of interest.

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
