# Peer review of "Effects of a Complex Physical Activity Program on Children’s Arithmetic Problem Solving and Arithmetic Reasoning Abilities"

_ejihpe, doi:10.3390/ejihpe13010010_

Round 1
Reviewer 1 Report
Dear Authors,
The work is aimed to investigate the effects of a 12-week Complex Physical Activity (CPA) intervention program on children's math performance. I find your work well structured and presented, however the research design needs clarification.
My comments and suggestions are listed point by point here and in the attached file.
Minor
Line 86: please avoid informal terminology, use mathematical instead of math.
Line 94: specify that also the easy-to-perform activity program was a physical activity.
Lines 99 and 126: specify if the intervention protocol of physical activity was held instead of the curricular PA lessons or in addition to them.
Major
The motor tests administered in the work were aimed to estimate aerobic and muscular fitness. While the intervention protocol was aimed to develop the students’ ability to control and adapt movements, focus their abilities, and to recall movements previously learned. In my opinion the selection of the motor tests does not reflect the motor abilities provided during the CPA. For this reason, some assumption proposed in the discussion and conclusion sessions are not supported by the present results. Authors must justify how the tests selected to evaluate the students’ physical fitness were relevant for the assessment of the physical abilities exercised in the intervention protocol.
Line 102 and onward: participants' descriptions should indicate whether any children with learning difficulties or disabilities was included. If this is the case, it should be specified if and how they were randomly assigned to the two groups.
Lines 127-128: “Standardized motor assessment tests and Math performance were assessed before and after the intervention period”, it would be appropriate to describe how long before and after the intervention period the tests were administered.
Line 148: “running speed was gradually increased with 1-minute intervals” does it mean that speed was increased every minute or that there was a minute of rest interval between each increase of speed?
Lines 150-151, 158-160 and 163-164: for each fitness test the test-retest reliability is reported, but there is no mention of the reliability protocols. In my opinion it would be helpful to include a description of how the tests’ reliability was assessed.
Table 1: all CPA exercises are listed, but number of repetitions and work/rest ratios are not included. In a 12 weeks intervention program an increase of exercise intensity and/or difficulty is expected. For both experimental and control groups, authors should provide detailed information about the training programs. Also, more details should be specified on the amount of simple, and easy-to-perform activity programs.
Lines 189-190: as in my previous comments on the tests selected to evaluate the students’ physical fitness the following assumption is not supported by the present results. “We believe that the CPA program proposed in this study helped to develop students' motor skills,”.
Lines 315-317: for the same reason the following assumption is not supported by the present results. “… sacrificing physical education classes for other curricular subjects … would lead to children having poor motor and cognitive skills.”

Reviewer 2 Report
I enjoyed reading your manuscript. Your work is of great value. It is not easy work. This all takes time. Please take my comments as positive additions to your work. Thank you.
Abstract, please provide Cohen’s d after each math test and physical functioning. The p < .001 is impressive. Knowing the meaningfulness is important.
Can you justify the 12-week CPA program? I know school schedules dictate 12-week programs or do at times. I believe some literature to indicate 12-weeks is suitable for fitness changes is required in your introduction section.
Line 97, small point – I am not sure Physical Education teachers needs to be capitalized.
Line 107, how did you determine if the student was relatively healthy? Did you exclude children with disabilities?
Line 127, not sure Math needs to be capitalized.
Small point, I am unsure of you use of numbers and then spelling the numbers out. It seems it should be two specific sections at line 168 and so on throughout your manuscript.
Table 1, I believe it will be easier to read with the exercises being left justified as opposed to centered. The – are not needed.
Line 214, same small point. I do not see the reason for Math to be capitalized.
I believe a figure of the Cohen’s d values will improve the manuscript. I know that the statistics are readable in the paragraphs. A figure will help someone browsing the article. This is just a suggestion.
I would merge the two conclusion paragraphs.
Section 3.1 and 3.2, as you defined the effect size interpretations in your methodology, I am not sure you need to in these sections. Again, a figure might be better.
Line 298, I do not think you need to place (n=128) here. Small sample size is enough. However, you did perform a power analysis. It is not as if you came up short.
Round 2
Reviewer 1 Report
Dear Authors,
I congratulate you on your work, even though you did not agree with some of my comments, the manuscript can be accepted in the present form.